# Evaluating the effectiveness and cost-effectiveness of health facility-based and community-based index-linked HIV testing strategies for children: protocol for the B-GAP study in Zimbabwe

Chido Dziva Chikwari,[1,2] Victoria Simms,[3] Stefanie Dringus,[4] Katharina Kranzer,[2,4] Tsitsi Bandason,[2] Arthi Vasantharoopan,[4] Rudo Chikodzore,[5] Edwin Sibanda,[6] Miriam Mutseta,[7] Karen Webb,[8] Barbara Engelsmann,[8] Gertrude Ncube,[9] Hilda Mujuru,[10] Tsitsi Apollo,[9] Helen Anne Weiss,[3] Rashida Ferrand[1,2]

**Correspondence to**
Chido Dziva Chikwari;
chido.dzivachikwari@lshtm.ac.uk

## ABSTRACT

**Introduction** The number of new paediatric infections per year has declined in sub-Saharan Africa due to prevention-of-mother-to-child HIV transmission programmes; many children and adolescents living with HIV remain undiagnosed. In this protocol paper, we describe the methodology for evaluating an index-linked HIV testing approach for children aged 2–18 years in health facility and community settings in Zimbabwe.

**Methods and analysis** Individuals attending for HIV care at selected primary healthcare clinics (PHCs) will be asked if they have any children aged 2–18 years in their households who have not been tested for HIV. Three options for HIV testing for these children will be offered: testing at the PHC; home-based testing performed by community workers; or an oral mucosal HIV test given to the caregiver to test the children at home. All eligible children will be followed-up to ascertain whether HIV testing occurred. For those who did not test, reasons will be determined, and for those who tested, the HIV test result will be recorded. The primary outcome will be uptake of HIV testing. The secondary outcomes will be preferred HIV testing method, HIV yield, prevalence and proportion of those testing positive linking to care and having an undetectable viral load at 12 months. HIV test results will be stratified by sex and age group, and factors associated with uptake of HIV testing and choice of HIV testing method will be investigated.

**Ethics and dissemination** Ethical approval for this study was granted by the Medical Research Council of Zimbabwe, the London School of Hygiene and Tropical Medicine and the Institutional Review Board of the Biomedical Research and Training Institute. Study results will be presented at national policy meetings and national and international research conferences. Results will also be published in international peer-reviewed scientific journals and disseminated to study communities at the end of study.

### Strengths and limitations of this study

► Our study will provide evidence for the effectiveness of index-linked HIV testing in facilities and communities. The strategy has the potential to be a cost-effective and efficient strategy for HIV testing in children, given the relatively low prevalence in this age group.

► Our intervention is relevant to policy questions for the implementation of HIV testing for children. This study will provide evidence for use of lower level cadres to offer and perform HIV testing as advocated by the WHO in both rural and urban settings.

► Other HIV testing interventions will be implemented by other stakeholders in the study areas during the study period. This may affect the impact assessment of this study.

► Potential challenges include low uptake of the intervention, lost-to-follow-up of clients once they have agreed to have their child tested and finding clients who have opted for home-based testing.

► Due to relatively low HIV prevalence (<4%), analysis of linkage to care will be limited to a purely descriptive analysis of the proportion of children linking to care and virologically suppressed at 12 months.

## INTRODUCTION

HIV testing is the critical first step to accessing life-saving antiretroviral therapy (ART).[1] Despite this, many children living with HIV in sub-Saharan Africa, which was home to approximately 92% of the 3.2 million children below 15 years with HIV globally in 2014, remain undiagnosed.[2–4]

Children infected through mother-to-child transmission who are not diagnosed in infancy often remain undiagnosed until

they present with symptomatic HIV infection in later childhood.[5] Coverage of early infant diagnosis is suboptimal and therefore a substantial proportion of children infected through mother-to-child transmission are not diagnosed timely in infancy and are only identified in later childhood when they develop advanced disease.[6] In 2017, only 63% of HIV-exposed infants in eastern and southern Africa received an HIV test within 8 weeks of age as recommended.[6] In a study that implemented provider-initiated HIV testing and counselling (PITC) among children aged 6–16 years in primary care clinics in Zimbabwe from 2013 to 2015, the median age of HIV diagnosis was 11 years and those aged >13 years were less likely than younger children to be diagnosed.[7 8] The high proportion of undiagnosed HIV among children in sub-Saharan Africa, together with low paediatric treatment coverage, continues to be key in sustaining the epidemic.

Barriers to HIV testing of children and adolescents include the need for guardian consent in most countries in sub-Saharan Africa for children younger than 16 years, negative attitudes and lack of training among healthcare workers, stigma and long distances between homes and healthcare facilities where testing is provided, inconvenient opening hours of facilities and long waiting times.[1]

A 2011 study from Zambia assessing reasons for non-uptake of HIV testing among children aged below 15 years found that the majority (76%) with confirmed or suspected HIV infection had a primary caregiver who was also living with HIV.[9] A study from Zimbabwe testing children aged 6–16 years conducted in 2014 found that 95% of those who tested HIV positive were perinatally infected. More importantly, 65% had a caregiver or sibling known to be HIV positive or taking ART, and 20% of the accompanying caregivers also tested HIV positive.[10] Similarly, in a Malawian study including patients (15–49 years) on ART at a large ART clinic in 2006–2007, 81% of their children (0–16 years) had reportedly not been tested.[11] These studies show that children living with HIV-infected adults are at high risk for being HIV positive themselves; however, in most instances, these children have not been provided with HIV testing.

The WHO has recommended index-linked HIV testing, whereby household members, sexual contacts or children of a known HIV-infected person are offered an HIV test, as one of the approaches for addressing the gap in HIV testing.[12] Two studies that have implemented such an approach among children in Malawi and Kenya found that more than 40% of adults in HIV care were living in households with children of unknown HIV status and the prevalence of HIV in children tested through this approach was significantly higher than from other 'unselective' HIV testing approaches within Africa.[13 14]

The Bridging the GAP in HIV testing and care for Children in Zimbabwe (B-GAP) study aims to investigate the effectiveness of index-linked HIV testing among children aged 2–18 years in Zimbabwe. Testing will use both facility-based and community-based approaches to address the barriers described above. Notably, as well as offering facility-based and home-based HIV testing for untested children of index adults with HIV, this study will also investigate a novel approach whereby caregivers can test their children using an oral mucosal HIV test. We elaborate on the formative research that informed the approaches to index-linked HIV testing, study design and methodology, analysis and intended study impact. In addition, the study will also evaluate the cost-effectiveness and conduct a process evaluation to inform factors that will influence scalability of this approach in programmatic settings. This study speaks strongly to the WHO recommendations to develop innovative feasible strategies to reduce the burden of undiagnosed HIV among children.[12]

## METHODS
### Study objectives
The aim of the study was to evaluate the effectiveness of index-linked HIV testing in identifying children with HIV aged 2–18 years in Zimbabwe. The objectives are to
1. Evaluate the acceptability, uptake and yield of an index-linked HIV testing strategy for children and adolescents in facility-based and community-based settings.
2. Investigate factors associated with uptake of index-linked HIV testing and with choice of a certain method (clinic-based, community-based provider-delivered and community-based caregiver-delivered testing).
3. Investigate linkage to care, retention in care and virological suppression among children living with HIV who are offered community health worker (CHW) delivered support in addition to clinic-based care (the standard of care).
4. Estimate the cost and calculate the cost-effectiveness of index-linked HIV testing for children and adolescents, compared with current standard of care.
5. Conduct a process evaluation of the intervention's implementation, mechanisms of impact and local context to inform the components required for sustainability and scalability.

### Patient and public involvement
#### Formative research and key considerations for study design
Between May and December 2017, we conducted extensive stakeholder engagement and a situational analysis to inform the intervention design with regard to the testing strategy. The specific objectives were first to understand HIV testing pathways within public healthcare facilities and their partnerships with non-governmental stakeholders; to inform selection of study sites and healthcare worker cadres to implement the intervention; and second to elicit key stakeholder views on preferences, potential limitations and foreseeable obstacles for delivery of index-linked HIV testing. Results were used to inform the intervention design and preparedness for implementation, as well as development of standard operating procedures.

**Table 1** Summary of how formative research informed intervention design

| Finding | Intervention design |
| --- | --- |
| Prevalence of undiagnosed HIV among children is heterogeneous across settings and regions. | Study sites were chosen based on an anticipated high prevalence of undiagnosed HIV. |
| Index-linked testing has not been fully implemented due to staff shortage at health facilities. | Research staff were hired to support implementation of intervention in facilities. |
| HIV testing services are provided by multiple stakeholders across the country with poor coordination between stakeholders. This results in duplication of services and failure to link individuals accessing services through community partners with health facilities. | A collaborative agreement between the research team and the MoHCC and its implementing partners conducting HIV testing in facilities and communities was established. HIV test kits will be provided by the MoHCC and all HIV testing data reported to the MoHCC. |
| User fees are in place for health service provision at the urban facilities including HIV testing (US$5 and US$3 depending on age of child). | For the purpose of this study, user fees will be dropped for all children and adolescents undergoing HIV testing in selected healthcare facilities |
| Individuals have to travel long distances to access healthcare facilities particularly in rural settings. | A novel HIV testing strategy will be introduced whereby caregivers will be given the option to test their children at home using an HIV self-test kit, thus eliminating the requirement for caregivers to bring children to the healthcare facility for testing. |

MoHCC, Ministry of Health and Child Care.

Interviews were conducted with Ministry of Health and Child Care (MoHCC) officials and key informants from MoHCC partner organisations that were implementing HIV programmes in Zimbabwe. In total, 25 field observations, 19 site assessments at healthcare facilities across 2 provinces and 53 in-depth interviews (IDIs) with healthcare providers, community-based organisations, adults and adolescents living with HIV and CHWs were undertaken. These findings will be reported in detail elsewhere; however, a summary of how findings informed design is shown in table 1.

Formative work was also conducted to inform the community-based support intervention for children living with HIV delivered by CHW. The research findings, development and design of the community-based support intervention will be presented in a separate manuscript.

### Study sites

Zimbabwe has the sixth highest adult HIV prevalence globally and approximately 1.3 million people were living with HIV in 2017.[6 15] The study will be conducted in six primary healthcare clinics (PHCs) in Bulawayo (urban) and three PHCs in Mangwe district (rural) in Zimbabwe. In 2016, Bulawayo had the highest adult HIV prevalence (18.7%) in an urban setting, while Mangwe in Matabeleland South Province has the highest national HIV prevalence (22.3%).[15] Clinics were purposively selected based on (1) the size of the facility so that the target sample size could be reached, and (2) their geographical locations to allow good accessibility by the study team in rural sites (figure 1). HIV clinics in hospitals were excluded due to their larger and less well-defined catchment area which would make community-based testing and follow-up difficult.

### Participant recruitment: inclusion and exclusion criteria

All individuals with HIV attending for care at study clinics will be screened daily to identify attendees with children aged 2–18 living in their household (index). If the index attendee is <18 years, they will need to be accompanied by a parent or caregiver aged ≥18 years to provide consent. Indices who have children with unknown status or children who previously tested HIV negative more than 6 months ago will be offered HIV testing for their children. Being tested more than 6 months ago was included as a criterion for offering testing to account for older children who may have been horizontally infected. If the index consents to having the child or children in the household tested, three testing options will be offered: (1) testing at the clinic by a healthcare provider, (2) home-based testing performed by a healthcare provider or (3) at home by the caregiver using a self-test kit.

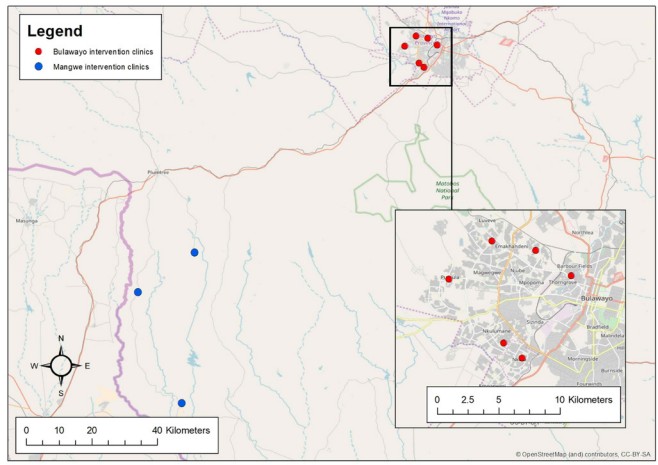

**Figure 1** Map of the selected Bridging the GAP in HIV testing and care for children in Zimbabwe sites.

Demographic details of the index and the children will be recorded electronically on tablets by the research assistants using Open Data Kit.[16] Locator details of the household including mobile phone numbers and physical addresses will be collected for all consenting index cases and particularly for community-based HIV testing and to ascertain HIV test outcomes among those who opt to test their children using a self-test kit. All participants will be given a study helpline number to call for further information, counselling referrals or support.

## HIV testing procedures

HIV testing will be carried out according to national guidelines using a serial testing algorithm.[17]

### Clinic-based HIV testing

Caregivers who opt for clinic-based HIV testing of their children will be given referral cards on the day of screening by a research assistant. HIV testing will be performed by routine clinic staff. Research assistants will use locator information to follow-up participants who have consented to have their child(ren) tested in the clinic but have not presented to the clinic within 7 days of recruitment. They will make up to three attempts to locate the index via telephone or home visit at day 7, 14 and 21 postscreening.

### Community-based HIV testing by a healthcare provider

Caregivers who opt for community-based testing will indicate a suitable date for HIV testing on the day of screening. A MoHCC implementing partner for community-based testing will visit the household to conduct an HIV test. If no contact is made on the scheduled visit date, the provider will conduct a further two visits to perform HIV testing. A test outcome will be recorded once three attempts have been made within 30 days of screening.

### Community-based HIV testing by the caregiver

Caregivers who opt to use an HIV oral mucosal test (OMT) to test their children at home will undergo an assessment of guardianship status for each child. Only indices who are parents or legal guardians of eligible children will be eligible to take an OMT kit. Detailed instructions on performing an oral HIV test and interpreting the test result will be given by research assistants in the clinic using an OMT, an instruction pamphlet from the manufacturer translated to local languages and demonstration videos. The research assistant will complete a brief competency assessment including a demonstration of how to perform the test and how to interpret a set of results displayed on pictures by the caregiver. If the caregiver fails the competency assessment, they will be asked to take up either of the other two testing options. Caregivers who successfully complete the competency assessment will be provided with a test kit for each eligible child and will be instructed to conduct the test within 5 days of screening and to keep the used OMT kit. They will be instructed to bring each child who tests OMT reactive for HIV to the clinic for confirmatory HIV testing as recommended by WHO guidelines.[18] Written information about confirmatory HIV testing services will be provided. Caregivers will be followed up to ascertain HIV test outcome either by telephone call or home visit within 7 days and their used/unused OMT test kits will be collected within 21 days of screening.

The first 10–15 index cases who select caregiver testing in each clinic will undergo supervised caregiver self-testing to evaluate if there are any problems with caregivers' understanding. The parent/guardian will conduct the test while a research assistant is present at the home on the selected testing date and in parallel the research assistant will conduct a rapid blood test as per the national HIV testing algorithm.[17] During the assessment, the research assistant will note any challenges the caregiver had with completing the test and whether or not the caregiver requested assistance from the research assistant. Appropriate changes will be made to the training and demonstration practices if necessary.

### Linkage to care if HIV positive

Children and adolescents will be told their HIV status according to their level of understanding and maturity as described by the national HIV testing and counselling guidelines.[17] All caregivers and children who test HIV positive will receive post-test counselling and a written referral to their nearest healthcare facility for confirmatory HIV testing and linkage to HIV care. For clinic-based testing, referrals will be made to the clinic staff responsible for initiating care on the day of testing. For community-based testing and caregiver testing, written referrals will be made to the nearest/preferred healthcare facility and all participants will be followed up by study staff to ascertain linkage to HIV care. Study staff will share all HIV testing records with the relevant healthcare facility. Caregivers of and children who test HIV positive (or are identified as known HIV positive but not linked to care) will be offered support visits from a CHW at home or at a community-based location of their choice conducted at 1, 3 and 6 months postdiagnosis.

In addition to the above-described testing procedures, partners of index cases will be offered HIV testing. This will not contribute to study outcomes but will be implemented to ensure the intervention complies with current standard of care and WHO recommendations.

## Outcomes

The primary outcome will be uptake of HIV testing, defined as an eligible child having completed an HIV test and the caregiver knowing the test result within 30 days of the HIV test being offered. The HIV test result or reasons why a test is not done will be recorded. Secondary outcomes for the study will include the preferred HIV testing method, HIV yield and prevalence. HIV prevalence and yield will be stratified by age group (0–5, 6–10 and 10–18 years). Additional secondary outcomes will be linkage to HIV care and viral load suppression at 12 months postdiagnosis.

## Cost-effectiveness analysis

The cost-effectiveness analysis aims to determine whether index-linked testing in children is a cost-effective strategy compared with current standard of care in Zimbabwe; passive provider-initiated testing at healthcare facilities. Both the full costs of delivering the intervention and the incremental costs will be estimated using a provider perspective through mixed methods involving a combination of primarily bottom-up micro-costing, with the use of top-down costing when necessary. Costs will be presented as follows: (1) Cost per child tested via each of the index-linked testing modalities; (2) Cost per HIV-positive child detected through each of the index-linked testing modalities; and (3) cost per new HIV initiate. The base case Cost-Effectiveness Analysis (CEA) will be conducted over a lifetime time horizon, annually discounted at the standard rate of 3%,[19] to reflect future costs and effects at current value, to estimate the incremental cost effectiveness ratio, as cost per disability-adjusted life year (DALY) averted. A dynamic transmission model will be used to estimate the DALYs averted, derived from a combination of accessing treatment and the HIV infections averted, using parameters extracted from the literature. A sensitivity analysis will be conducted to explore the impact of key variables on the results including HIV prevalence and geographical location. As there is much debate and uncertainty surrounding recommended C-E threshold values for different settings,[20] a cost-effectiveness acceptability curve will be constructed. The resulting Incremental Cost-Effectiveness Ratio (ICER) will be compared against different threshold values that are likely to be appropriate to the Zimbabwe setting. This analysis will help inform MoHCC policy around alternate forms of HIV testing and counselling.

## Process evaluation

A detailed mixed-methods process evaluation will be conducted alongside the delivery of the intervention. The process evaluation will be based on the Medical Research Council Process Evaluation Framework and will explore three core evaluation functions: implementation, mechanisms of impact and context to evaluate the delivery and receipt of the intervention, in order to better understand what components of the intervention work, for whom and under what circumstances.[21] The process evaluation data will be analysed and written up in order to contribute to understanding the study results and to provide guidance for sustainable and scalable implementation of the intervention by local health authorities should it prove successful. Quantitative indicators for the process evaluation will include staff turnover, record of shortage of test kits and supplies and incident reports about facility-based and community-based testing from research assistants and the project coordinator. Descriptive statistics of these indicators will be used to help assess aspects of the intervention's fidelity, feasibility and acceptability, and to identify lessons important in informing the replication and scale-up of this approach to testing.

Focus group discussions (FGDs) with research assistants and FGDs and IDIs with a purposively selected sample of indices will be conducted to elicit provider and caregiver perceptions and experiences of index-linked HIV testing for children at baseline and end line. Sampling for FGDs will include a group of caregivers representing index age (older vs younger) and sex, HIV testing option, HIV test uptake (accepted vs did not accept testing; and whether child tested or not) and study site (rural vs urban). We will also conduct IDIs with caregivers who had a child test HIV positive through index-linked HIV testing at end line.

As testing by caregivers is a novel intervention, the acceptability of this method will be explored from providers' and caregivers' perspectives. IDIs will be held with caregivers who selected this option to understand how messaging and training for caregiver testing used in the competency and accuracy assessments can be improved, as well as how testing is performed by caregivers on their children in their homes. Interviews will also be held with index participants who did not select his method to understand their concerns.

## Sample size estimates

Sample size estimations are based on precision of estimates of uptake of index-linked HIV testing. We will recruit participants from nine clinics, with an average daily adult attendance of 32 per clinic (excluding repeat attenders). Over 12 weeks per clinic, we expect to screen a total of 5184 HIV-positive index cases at nine clinics with children aged 2–18 years (assuming 5 working days a week and 30% of adults with HIV have children and adolescents living in the household). Assuming an index refusal rate of 35%, and an average of 2 children per household, we would screen approximately 6739 children and adolescents. This sample size provides ±1% precision for an estimate of 80% of screened children and adolescents taking up HIV testing, with a 95% CI of 79.0%–81.0%. Sample sizes for qualitative work will be determined on an ongoing basis by study staff until thematic saturation has been reached.

Assuming HIV prevalence of 4%, this would yield 215 children living with HIV eligible for linkage-to-care. This sample size provides good precision of ±6% around an estimated 80% linking-to-care (n=172) and precision of ±9% around an estimated 70% with viral suppression at 12 months out of 85% retained in care (n=102/146).

## Data analysis

A study flowchart showing total number of index cases screened, eligibility of index cases and children in their household, proportions of index cases providing consent for collection of additional data about themselves and the children living in their households will be presented (figure 2). Demographic characteristics of index participants and children and adolescents living in their households will be summarised using proportions, means and SD. The uptake of HIV testing and HIV prevalence stratified by age group (0–5, 6–10 and 10–18 years) and sex will

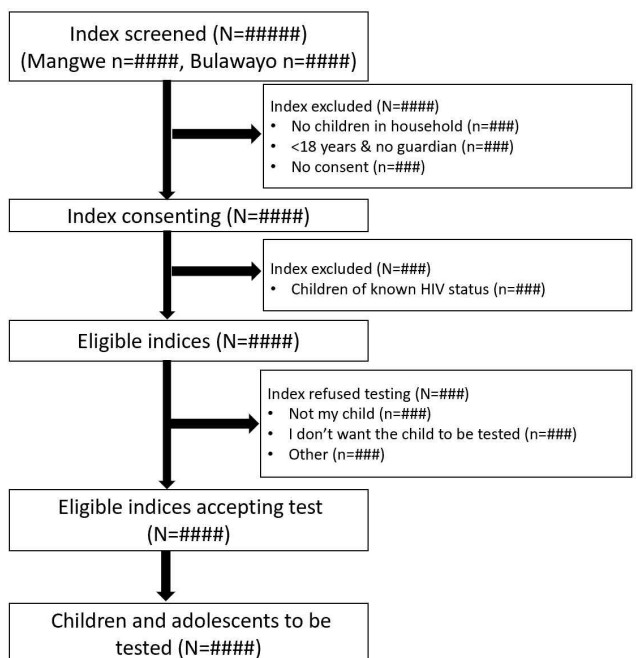

**Figure 2** Index-linked HIV testing participant flow.

be calculated, and factors associated with uptake of index-linked HIV testing will be assessed using univariate analysis. A multivariate logistic regression will be conducted to assess factors predictive of uptake/non-uptake of HIV testing adjusted for clustering by index case. We will evaluate factors associated with the uptake of the preferred HIV testing method using multinomial logistic regression. The proportion of children who tested positive and linked to HIV care and the proportion with virological suppression will be calculated at 12 months.

### Study status

The study began recruitment of study participants in January 2018. Recruitment is ongoing.

### DISCUSSION

There has been a remarkable scale-up of prevention of mother-to-child HIV transmission (PMTCT) programmes resulting in a decline in numbers of perinatally acquired HIV infections.[2] However, many children and adolescents with HIV, acquired before PMTCT programmes were scaled-up, remain undiagnosed, with diagnosis only occurring when they present with advanced disease.[6] In addition, suboptimal coverage of early infant diagnosis within PMTCT programmes results in many HIV-exposed children not being identified.[2 6] Those who acquire HIV postnatally through breastfeeding may also present later in childhood. HIV testing is the critical step to accessing HIV treatment and high rates of undiagnosed HIV partly explain the lower coverage of HIV treatment in children and adolescents compared with adults.[22]

PITC has been recommended by the WHO since 2007.[23] However, this relies on an individual attending a health facility. Also, PITC may not be as cost-effective

a strategy in children as in adults given the lower HIV prevalence among children compared with adults. Index-linked testing is a targeted strategy focusing on testing of children at higher risk of HIV infection. If effective, this strategy has the potential to be a more cost-effective and efficient HIV testing strategy in resource-limited settings. Our study will implement both community-based and health facility-based HIV testing for children and investigate a novel approach of caregivers testing their own children to increase accessibility and uptake of HIV testing.

We believe that community-based strategies in our study have the potential to eliminate many of the barriers to HIV testing for children and adolescents.[14] In a study conducted in Malawi, index-linked testing offered to children and young people (1–24 years) had higher uptake for community (95.5%) when compared with facility-based (4.5%) testing.[14] Enabling caregivers to test their children may reduce the burden of client-associated costs such as time spent in and getting to facilities for testing by clients, provider time, and maintenance and upkeep of health facilities. However, most importantly, it allows the caregivers to perform the test in the comfort and privacy of their homes and empowers them to take responsibility for the testing. Another innovation is the use of CHWs to perform HIV testing in the community. While a similar cadre termed 'primary care counsellor' is widely used to provide HIV testing in health facilities, HIV testing in communities is largely provided by nurses. The CHWs will complete a 2-week training in ethics and rapid HIV testing as well as on study procedures. If effective, our study will provide evidence for use of lower level cadres such as community health workers to offer and perform HIV testing as advocated by the WHO.[12] There is evidence to show that the use of lower level cadres can reduce cost of staffing for the government and may improve cost-effectiveness of community-based approaches when compared with traditional facility-based HIV testing.[24]

Furthermore, the study also addresses linkage to care, initiation of ART and adherence through a community-based support intervention. It is crucial that any HIV testing intervention addresses these steps in the continuum of HIV care to ensure maximum benefits (ie, reduced morbidity and onward transmission).

Potential challenges include low uptake of the intervention, lost-to-follow-up of clients once they have agreed to have their child tested and finding clients who have opted for home-based testing. Within our study, user fees for HIV testing were dropped. This has implications for the generalisability of our findings in settings where user fees are in place and may act as a barrier to uptake of HIV testing services. Similarly, our study findings may not be generalisable to other settings as contextual factors may affect uptake of the intervention. It is important to note that in parallel to evaluation of uptake and yield of index-linked HIV testing, we will conduct a detailed process evaluation to explore the implementation, mechanisms of impact and contextual factors affecting the delivery and acceptability of this intervention, which will help

inform sustainability and scale-up should it prove effective. Another limitation is that other HIV testing interventions will be implemented in our study settings at the same time. However, detailed recording of other HIV testing interventions will be part of the process evaluation. This will help with understanding how other HIV testing interventions interact with each other and index-linked testing.

### Ethics and dissemination

Ethical approval was sought and granted by the Medical Research Council of Zimbabwe (MRCZ), the London School of Hygiene and Tropical Medicine ethics committee and the Institutional Review Board of the Biomedical Research and Training Institute.

Study progress and findings will be reported annually to MRCZ and feedback meetings will be held quarterly with the health directorates of the two study sites. Results of interim data analysis will be presented at national and international research meetings and conferences. Results will also be prepared for publication in international peer-reviewed scientific journals and disseminated to study communities at the end of study. A detailed dissemination plan to inform scale-up will be developed for implementation.

#### Author affiliations

[1]Department of Clinical Research, London School of Hygiene and Tropical Medicine, London, UK
[2]Biomedical Research and Training Institute, Harare, Zimbabwe
[3]MRC Tropical Epidemiology Group, London School of Hygiene and Tropical Medicine, London, UK
[4]London School of Hygiene and Tropical Medicine, London, UK
[5]Matebeleland South, Ministry of Health and Child Care, Bulawayo, Zimbabwe
[6]City Health Department, Bulawayo City Council, Bulawayo, Zimbabwe
[7]Population Services International Zimbabwe, Harare, Zimbabwe
[8]Organization for Public Health Interventions and Development, Harare, Zimbabwe
[9]Ministry of Health and Child Care Zimbabwe, Harare, Zimbabwe
[10]College of Health Sciences, University of Zimbabwe, Harare, Zimbabwe

**Contributors** RF conceptualised the project. CDC developed the study protocol and data collection tools. AV and VS contributed to the design of cost-effectiveness evaluation of the study. SD, RF and CDC designed the study process evaluation. RC, MM, BE, KW, TB and ES contributed to the formative work and development of project logistics. HAW, KK, GN, HM and TA provided technical input to the design and statistical analysis of the study. All authors have read and approved the final manuscript.

**Funding** This study is jointly funded by the UK Medical Research Council (MRC) and the UK Department for International Development (DFID) under the MRC/DFID Concordat agreement and is also part of the EDCTP2 programme supported by the European Union grant (MR/P011268/1). RF is funded by the Wellcome Trust through a Senior Fellowship in Clinical Science (206316/Z/17/Z).

**Map disclaimer** The depiction of boundaries on the map(s) in this article do not imply the expression of any opinion whatsoever on the part of BMJ (or any member of its group) concerning the legal status of any country, territory, jurisdiction or area or of its authorities. The map(s) are provided without any warranty of any kind, either express or implied.

**Competing interests** None declared.

**Patient consent for publication** Not required.

**Provenance and peer review** Not commissioned; externally peer reviewed.

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
