## [Reviewer comments · BMJ Open]

ARTICLE DETAILS

TITLE (PROVISIONAL)	Evaluating the effectiveness and cost-effectiveness of health facility- and community-based index-linked HIV testing strategies for children: protocol for the B-GAP study in Zimbabwe
AUTHORS	Dziva Chikwari, Chido; Simms, Victoria; Dringus, Stefanie; Kranzer, Katharina; Bandason, Tsitsi; Vasantharoopan, Arthi; Chikodzore, Rudo; Sibanda, Edwin; Mutseta, Miriam; Webb, Karen; Engelsmann, Barbara; Ncube, Gertrude; Mujuru, Hilda; Apollo, Tsitsi; Weiss, Helen; Ferrand, Rashida

VERSION 1 - REVIEW

REVIEWER	Benjamin Johns Abt Associates, Inc. USA
REVIEW RETURNED	20-Feb-2019

GENERAL COMMENTS	Overall, this is a well-written and well thought out paper worthy of publication. I have a few minor considerations for the authors to consider: Page 6 Lines 24-25 "These findings will be reported in detail elsewhere" - update with a reference if available. Page 10 Lines 47-48: It looks like the authors should consider multivariate multinomial logistic regression rather than or in addition to multivariate logistic regression since they are proposing to look at the preferred HIV testing method. Page 9 Paragraph on cost-effectiveness analysis: Three thoughts about this paragraph. First, the authors do not specify how the results will be used. It would be beneficial to define at this point what will be considered a cost-effective result - below what cost per DALY averted will be considered cost-effective? (e.g., recent literature is more strongly suggesting that GDP per capita is not a good threshold). Second, it is not clear from this write up how valid or how at all the authors will generalize the results of the evaluation (which are deliberately set in high prevalence settings, etc.) to 'a nationwide sweep of children'. One or two sentences describing what adjustments will need to be made to the evaluation results would be helpful. Third, the phrase 'decision analytic modeling' is overly generic. I think at this point the authors can/should provide more specificity as to the type of modeling they will employ (decision tree? cohort Markov model? etc.).
---

REVIEWER	Brooke Nichols Department of Global Health, Boston University, United States
REVIEW RETURNED	22-Feb-2019

GENERAL COMMENTS	Overall, this study is of great importance as it is a patient-centric intervention aimed at increasing uptake of HIV testing among hard to reach populations such as children and adolescents, identify vulnerable children who may be at risk of contracting HIV and identify those infected who may have been missed by sub-optimal routine early infant diagnosis. There are some parts of the protocol, however, that could use additional detail. Major revisions:  1. Page 7, line 20: Will the date of diagnosis of the index client be recorded or used in any way? I can imagine that if the index client has a 10 year old, and the index client was diagnosed 6 months ago (with, say, a CD4 cell count of 600), that the 10 year old was probably not at risk of vertical transmission. 2. Page 8, line 52: "All children who test HIV positive will receive post-test counselling, and a written referral to their nearest healthcare facility for confirmatory HIV testing and linkage to HIV care." Will there be any differentiated approach based on the age of the child? Since this is under the section 'community based HIV testing by the caregiver' – is the caregiver giving a written referral to the nearest healthcare facility to the child? That seems a bit odd and could use clarification. 3. Page 9, line 21-32, Cost-effectiveness analysis: The CEA is comparing index testing against SOC, however index testing has three testing modalities which subsequently will have different costs and outcomes. Please mention whether each of the index testing modalities will be evaluated individually evaluated against the SOC, as a group, or if there will be an incremental cost analysis. Additionally, how will the DALYs be calculated or estimated? Directly from the study, or through the literature? You may also want to consider a concrete cost-outcomes analysis, such as cost per new ART initiate- this may be easier to interpret. 4. Page 9, line 36, Process evaluation: The authors plan to collect a number quantitative indicators for this analysis. It would be useful to outline how these indicators will be assessed interpreted in order to determine/inform scalability and sustainability of the intervention. Minor revisions:  1. Page 5, line 7: Is B-GAP an acronym? If so please provide the full name 2. Page 11, line 35: 'Enabling caregivers to test their children may reduce the burden of provider associated costs such as time spent in facilities for testing by clients,...' Time spent waiting in facilities by clients is not a provider cost, but rather a cost to the patient.
--

REVIEWER	Michael Irvine University of British Columbia, Canada
REVIEW RETURNED	06-Mar-2019

GENERAL COMMENTS	Thank you for the opportunity to review the manuscript "Evaluating the effectiveness and cost-effectiveness of health facility- and community-based index-linked HIV testing strategies for children: protocol for the B-Gap study in Zimbabwe". Overall, I found the
---

	protocol and study description to be comprehensive and well-explained. I only have a couple of main comments on areas that could be expanded upon. Main Comments Under Table 1, it is indicated that costs for testing will be dropped for purpose of study. Can authors speak to the potential limitation this may place on generalizing the results of the study if costs remain in place post-study? Costs would likely create a barrier to testing, which may hinder the further effectiveness of the intervention. I would like to see more details be provided on the under the cost-effectiveness analysis. For the decision analytic model, will secondary cases averted be factored in or will only primary cases be considered? Perhaps a brief outline of the proposed model will be helpful here. Can the authors also provide the time-horizon that will be considered to determine cost-effectiveness. Also, it would be beneficial to specify the criterion to determine cost-effectiveness i.e. what is the incremental cost-effectiveness ratio threshold defined to be cost-effective. Minor comments Abstract line 22: Clarify them is referring to children of caregiver. Abstract line 24: "HIV test result will be recorded." Page 10 line 54: FDG -> FGD
--	--

VERSION 1 – AUTHOR RESPONSE

Reviewer(s)' Comments to Author:

Reviewer: 1

Reviewer Name: Benjamin Johns

Institution and Country: Abt Associates, Inc. USA

Overall, this is a well-written and well thought out paper worthy of publication. I have a few minor considerations for the authors to consider:

- 1) Page 6 Lines 24-25 "These findings will be reported in detail elsewhere" - update with a reference if available.

Thank you for your comment. Unfortunately, this data is still in draft format and therefore we are unable to provide a reference at present.

2) Page 10 Lines 47-48: It looks like the authors should consider multivariate multinomial logistic regression rather than or in addition to multivariate logistic regression since they are proposing to look at the preferred HIV testing method.

Thank you, this is a useful comment. Test location choice is a critical component of the study and we are planning to evaluate the factors associated with choice of test location and have revised the text to include multinomial logistic regression. It is critical, however, to note that in some cases test location choice will be different to actual test location (e.g. those who selected clinic-based testing at screening may subsequently have children tested in the community at follow up). This will be detailed and evaluated in the main outcomes paper.

3) Page 9 Paragraph on cost-effectiveness analysis: Three thoughts about this paragraph. First, the authors do not specify how the results will be used. It would be beneficial to define at this point what will be considered a cost-effective result - below what cost per DALY averted will be considered cost-effective? (e.g., recent literature is more strongly suggesting that GDP per capita is not a good threshold). Second, it is not clear from this write up how valid or how at all the authors will generalize the results of the evaluation (which are deliberately set in high prevalence settings, etc.) to 'a nationwide sweep of children'. One or two sentences describing what adjustments will need to be made to the evaluation results would be helpful. Third, the phrase 'decision analytic modelling' is overly generic. I think at this point the authors can/should provide more specificity as to the type of modelling they will employ (decision tree? cohort Markov model? etc.).

Thank you for your comment. The goal of this analysis is to inform Ministry of Health and Child Care (MoHCC) policy with regard to alternate forms of HIV testing and counselling. As mentioned, given the debate and uncertainty surrounding existing and commonly used (WHO) C-E thresholds, coupled with the fact that specifying a precise threshold in advance is poor health economics etiquette, we propose constructing a cost-effectiveness acceptability curve and comparing against different threshold values appropriate to the Zimbabwe setting. Upon deliberation after receiving reviewer comments, we realized that modelling the scale-up of B-GAP nationwide is beyond the scope of this protocol paper and have omitted it. Regarding the modelling approach, we will be employing a dynamic transmission model. Given that an existing dynamic transmission model will have to be adapted for the adolescent population, a unique population which has not been modelled before, or newly created, we are currently in the process of conducting a review of the literature to inform the adaptation/development of a model based on current best practices.

Reviewer: 2

Reviewer Name: Brooke Nichols

Institution and Country: Department of Global Health, Boston University, United States

Overall, this study is of great importance as it is a patient-centric intervention aimed at increasing uptake of HIV testing among hard to reach populations such as children and adolescents, identify vulnerable children who may be at risk of contracting HIV and identify those infected who may have

been missed by sub-optimal routine early infant diagnosis. There are some parts of the protocol, however, that could use additional detail.

Major revisions:

1) Page 7, line 20: Will the date of diagnosis of the index client be recorded or used in any way? I can imagine that if the index client has a 10-year-old, and the index client was diagnosed 6 months ago (with, say, a CD4 cell count of 600), that the 10-year-old was probably not at risk of vertical transmission.

Thank you for this very critical observation. As part of the data collected during screening, we will also be collecting date of HIV diagnosis for the index. When designing the study protocol, we envisioned that the duration that an index has known their HIV status may have an impact on whether or not the children that live in their household have been tested already and may also affect uptake of testing i.e. if they were diagnosed 2 years ago they are more likely to have disclosed to their family and are therefore more likely to accept testing for their children. This is in contrast to an index who has been recently diagnosed and has not disclosed their HIV status to their family. As you correctly noted if the index is the biological mother of the child duration since diagnosis and subsequent CD4 cell count at the time may have a link to the risk of vertical HIV transmission.

We have not detailed all demographic data collected for the index and children as part of this protocol paper, however, data collection tools for the study will be published together with the final study data.

2) Page 8, line 52: "All children who test HIV positive will receive post-test counselling, and a written referral to their nearest healthcare facility for confirmatory HIV testing and linkage to HIV care." Will there be any differentiated approach based on the age of the child? Since this is under the section 'community-based HIV testing by the caregiver' – is the caregiver giving a written referral to the nearest healthcare facility to the child? That seems a bit odd and could use clarification.

Thank you for this insightful comment. We have now added a subheading to clarify that the steps to be followed for linkage to care apply to all the 3 test locations, however, have expanded on procedures for the two community-based testing methods. We have also added a sentence describing procedures for HIV status disclosure based on the level of understanding and maturity of the child as per the Zimbabwe national HIV testing and counselling guidelines.

3) Page 9, line 21-32, Cost-effectiveness analysis: The CEA is comparing index testing against SOC; however, index testing has three testing modalities which subsequently will have different costs and outcomes. Please mention whether each of the index testing modalities will be evaluated individually evaluated against the SOC, as a group, or if there will be an incremental cost analysis. Additionally, how will the DALYs be calculated or estimated? Directly from the study, or through the literature? You may also want to consider a concrete cost-outcomes analysis, such as cost per new ART initiate- this may be easier to interpret.

Thank you for your comment. Each of the three testing modalities will be evaluated individually, with both full and incremental costs being presented. Outcomes of interest for the cost analysis are as follows: 1.) Cost per child tested via each of the testing modalities; 2.) Cost per HIV positive child detected via each testing modality; 3.) Cost per new HIV initiate. DALYs will be estimated through the literature. We have now described this in detail within the manuscript.

4) Page 9, line 36, Process evaluation: The authors plan to collect a number of quantitative indicators for this analysis. It would be useful to outline how these indicators will be assessed and interpreted in order to determine/inform scalability and sustainability of the intervention.

Thank you for this comment and suggestion. We have now added text to the manuscript to describe how these indicators will be used; descriptive statistics of these indicators will be used to help assess aspects of the intervention's fidelity, feasibility and acceptability, and to identify lessons important in informing the replication and scale-up of this approach to testing.

Minor revisions:

1) Page 5, line 7: Is B-GAP an acronym? If so, please provide the full name

Thank you for this comment. We have now included the full name for the B-GAP study (Bridging the GAP in HIV testing and care for Children in Zimbabwe).

2) Page 11, line 35: 'Enabling caregivers to test their children may reduce the burden of provider associated costs such as time spent in facilities for testing by clients,' Time spent waiting in facilities by clients is not a provider cost, but rather a cost to the patient.

Thank you for this comment. We have now reworded this sentence to reflect that time spent in facilities is a cost to the client rather than the provider.

Reviewer: 3

Reviewer Name: Michael Irvine

Institution and Country: University of British Columbia, Canada

Thank you for the opportunity to review the manuscript “Evaluating the effectiveness and cost-effectiveness of health facility- and community-based index-linked HIV testing strategies for children: protocol for the B-Gap study in Zimbabwe”. Overall, I found the protocol and study description to be comprehensive and well-explained. I only have a couple of main comments on areas that could be expanded upon.

Main Comments

1) Under Table 1, it is indicated that costs for testing will be dropped for purpose of study. Can authors speak to the potential limitation this may place on generalizing the results of the study if costs remain in place post-study? Costs would likely create a barrier to testing, which may hinder the further effectiveness of the intervention.

Thank you for this comment. We have now added detail to our discussion highlighting the effect of removing user fees on the generalizability of our study findings.

2) I would like to see more details be provided on the under the cost-effectiveness analysis. For the decision analytic model, will secondary cases averted be factored in or will only primary cases be considered? Perhaps a brief outline of the proposed model will be helpful here. Can the authors also provide the time-horizon that will be considered to determine cost-effectiveness. Also, it would be beneficial to specify the criterion to determine cost-effectiveness i.e. what is the incremental cost-effectiveness ratio threshold defined to be cost-effective.

Thank you for your comment. Regarding the modelling approach, we will be employing a dynamic transmission model. Given that an existing dynamic transmission model will have to be adapted for the adolescent population, a unique population which has not been modelled before, or newly created, we are currently in the process of conducting a review of the literature to inform the adaptation/development of a model based on current best practices. The base case CEA will be modelled over a lifetime time horizon. Given the debate and uncertainty surrounding existing and commonly used (WHO) C-E thresholds, coupled with the fact that specifying a precise threshold in advance is poor health economics etiquette, we propose constructing a cost-effectiveness acceptability curve and comparing against different threshold values appropriate to the Zimbabwe setting.

Minor comments

1) Abstract line 22: Clarify them is referring to children of caregiver.

Thank you for this point of clarification. We have now noted in the text that it is the caregiver who will be testing the children.

2) Abstract line 24: “HIV test result will be recorded.”

Thank you. We have now corrected the text.

3) Page 10 line 54: FDG -> FGD

Thank you. We have now corrected the text.

VERSION 2 – REVIEW

REVIEWER	Benjamin Johns Abt Associates, Inc.
REVIEW RETURNED	01-May-2019

GENERAL COMMENTS	The authors have addressed the reviewers' concerns.
---

REVIEWER	Brooke Nichols Boston University, United States of America
REVIEW RETURNED	23-Apr-2019

GENERAL COMMENTS	The authors have thoroughly addressed all of my comments.
---

REVIEWER	Michael Irvine University of British Columbia, Canada
REVIEW RETURNED	07-May-2019

GENERAL COMMENTS	Thank you for the opportunity to review the revised version of this study protocol. I am pleased to say that all of my main comments have been addressed in this revision and I have no further concerns. I wish the authors the best of luck in the study and look forward to their findings.
---